# The Effect of Plasma Activated Water on Maize (*Zea mays* L.) under Arsenic Stress

**DOI:** 10.3390/plants10091899

**Published:** 2021-09-14

**Authors:** Zuzana Lukacova, Renata Svubova, Patricia Selvekova, Karol Hensel

**Affiliations:** 1Department of Plant Physiology, Faculty of Natural Sciences, Comenius University, Ilkovičova 6, 842 15 Bratislava, Slovakia; renata.subova@uniba.sk; 2Division of Environmental Physics, Faculty of Mathematics, Physics and Informatics, Comenius University, 842 48 Bratislava, Slovakia; ivanova.patricia@gmail.com (P.S.); karol.hensel@fmph.uniba.sk (K.H.)

**Keywords:** antioxidants, arsenic, maize, plasma activated water

## Abstract

Plasma activated water (PAW) is a source of various chemical species useful for plant growth, development, and stress response. In the present study, PAW was generated by a transient spark discharge (TS) operated in ambient air and used on maize corns and seedlings in the 3 day paper rolls cultivation followed by 10 day hydroponics cultivation. For 3 day cultivation, two pre-treatments were established, “priming PAW” and “rolls PAW”, with corns imbibed for 6 h in the PAW and then watered daily by fresh water and PAW, respectively. The roots and the shoot were then analyzed for guaiacol peroxidase (G-POX, POX) activity, root tissues for their lignification, and root cell walls for in situ POX activity. To evaluate the potential of PAW in the alleviation abiotic stress, ten randomly selected seedlings were hydroponically cultivated for the following 10 days in 0.5 Hoagland nutrient solutions with and without 150 μM As. The seedlings were then analyzed for POX and catalase (CAT) activities after As treatment, their leaves for photosynthetic pigments concentration, and leaves and roots for As concentration. The PAW improved the growth of the 3 day-old seedlings in terms of the root and the shoot length, while roots revealed accelerated endodermal development. After the following 10 day cultivation, roots from PAW pre-treatment were shorter and thinner but more branched than the control roots. The PAW also enhanced the POX activity immediately after the imbibition and in the 3 day old roots. After 10 day hydroponic cultivation, antioxidant response depended on the PAW pre-treatment. CAT activity was higher in As treatments compared to the corresponding PAW treatments, while POX activity was not obvious, and its elevated activity was found only in the priming PAW treatment. The PAW pre-treatment protected chlorophylls in the following treatments combined with As, while carotenoids increased in treatments despite PAW pre-treatment. Finally, the accumulation of As in the roots was not affected by PAW pre-treatment but increased in the leaves.

## 1. Introduction

Atmospheric plasma has shown promising potential in various agricultural applications, where it is applied to seeds or plants to stimulate germination, or to modulate growth, and fruit yield [1,2]. The plasma can be applied either directly or indirectly, i.e., its effect is mediated by a gas or a liquid exposed to plasma. The plasma produces various gaseous reactive species, which may dissolve into liquid/water to produce so-called plasma activated liquid or plasma activated water (PAW). The composition and activity of PAW can be tuned by various parameters, e.g., discharge type and its power, gas and water composition, and flow rate. An interesting feature of PAW, having potential also in commercial use, is that it contains various reactive oxygen and nitrogen species (RONS), such as nitrates (NO_3_^−^), nitrites (NO_2_^−^), and hydrogen peroxide (H_2_O_2_) [3], and is able to preserve its antibacterial activity for several days [4]. This feature predicts its use in agriculture where an increase in human population is reflected in a higher food demand. Nowadays, the use of chemicals to decrease bacterial contamination of seed surface, or the use of pesticides and herbicides to avoid pathogens and weeds, brings secondary contamination of soils and water. Additionally, the use of fertilizers, especially those with low quality and control, to increase the crop yield often results in the heavy metal soil contamination [5]. On the contrary, with the use of PAW or plasma activated ammonia solutions, such contamination can be avoided, as these solutions may serve as an effective source of nitrogen with nitrates (NO_3_^−^) and ammonium ions (NH_4_^+^) being the most important compounds for plant growth and development [6]. Further, hydrogen peroxide (H_2_O_2_) can serve to sterilize seeds and can also enhance seed priming. It may also act as a signaling molecule and activate proteins or genes responsible for plant growth and development. As a result of several reported positive effects of plasma use toward seed and plants, a new field has been established and is referred as “plasma agriculture” [1]. Atmospheric plasma and PAW in agriculture have been studied in recent years for their effects on seeds, mostly to improve their germination, growth, and subsequent yield [7,8,9,10]. They were also reported being able to change enzymatic activity in seeds [11,12,13,14,15], to alter secondary metabolites content [16,17], to induce structural modification of seed surface and associated changes in affinity towards water [18], and to reduce numbers of phytopathogenic microorganisms on the seed surface [19,20,21]. The effect of plasma and PAW have been also intensively studied for various plant species cultivated in different ways and analyzed by various methods. Their positive effects on macroscopic physical characteristics of seedlings and plants have been reported, including number and quality of leaves, length of above-ground parts and roots, and fresh and dry weight. The broad range of analytical methods have been used also to investigate the effects of plasma and PAW on physiological processes and metabolism, e.g., water uptake, photosynthetic pigments content, photosynthetic rate, enzymatic activity, and protein contents or DNA damage [22,23,24,25].

Plants are often challenged by various stresses. One of the most common is a stress from toxic elements contaminating soils. Among them, one of the most dangerous is metalloid arsenic (As), absorbed and translocated in the plant bodies into the edible parts where it threatens the highest trophic levels [26]. A common plant reaction to As stress is an overproduction of various reactive oxygen species (ROS) within the plant body [27] triggering antioxidant systems [28]. H_2_O_2_ plays a central role in stress signal transduction [27,29]; a delicate balance between its production and scavenging must be maintained in the plant cells. Too high level of ROS causes damage, especially to cell macromolecules, and this leads to cell death [30]. On the other side, the non-toxic H_2_O_2_ concentration, acting as signal molecules, activate multiple plant cell responses, especially via MAPK (mitogen-activated protein kinases) cascades, leading to ROS detoxification and surviving stress situation [29]. Systems joined with ROS in terms of their metabolism are either enzymatic or non-enzymatic. The most active enzymatic scavengers of H_2_O_2_ are catalase (CAT), decomposing it directly, and peroxidases (POX), reducing it by oxidizing various substrate, e.g., monolignols in the cell wall, which is an important step in lignin formation [28,31,32,33]. A promising method of inducing stress resistance in plants is the pre-treatment (priming) of seeds or plants by exposure to a chemical compound acting as a stressor [27,34,35,36]. Studies have revealed that priming phenomenon modulates the plant response positively to the followed-up stress. However, the molecular mechanism associated with priming is still to be elucidated, although there is evidence suggesting the role of agents such as H_2_O_2_ (and others) making plants more tolerant [27,37,38,39], especially by modulating ROS detoxification pathways [40,41].

The aim of the present study was to investigate the effect of PAW generated by a transient spark discharge (TS) operated in ambient air in contact with tap water on maize corns and young seedlings. Several treatments of PAW were established, and their potential in the priming of seedlings subsequently exposed to stress from arsenic (As) was evaluated. This was documented for the first time to our best knowledge and broadens the understanding of PAW interaction with the plant defensive systems. Maize seeds and seedlings were pre-treated in PAW for three days and subsequently analyzed for their POX and CAT activities, lignification of the root tissues, and in situ POX activity in roots. The pre-treatment was followed by 10 day hydroponic cultivation in 0.5 Hoagland nutrient solution with and without As stress. Subsequently, POX and CAT activity in seedlings, chlorophyll and carotenoid concentrations in leaves, and concentrations of As in leaves and roots were determined.

## 2. Results

The plasma activated water (PAW) produced in the present study had pH 7.5 and contained approximately 0.5 ± 0.1 mM of H_2_O_2_, 0.6 ± 0.1 mM of NO_2_^−^, and 1.7 ± 0.3 mM of NO_3_^−^. Exposure of water solutions, such as deionized water, or physiological solution to air plasmas usually leads to their acidification and a decrease in pH. However, the pH of tap water after plasma exposure remained fairly constant or changed very mildly due to its natural hydrocarbon buffer system. To investigate the effect of PAW containing these RONS, the growth parameters, antioxidant enzyme (G-POX, POX) activity, and the development of the young root of maize seedlings were assessed. First, activity of G-POX was measured in the corns imbibed in PAW. The enzymatic activity increased by more than four times in the maize corns after only 6 h imbibition in the PAW in comparison with corns in control (tap water) (Figure 1a). To investigate young seedlings exposed to PAW, corns were germinated and left growing in paper rolls for three days; the PAW treatment was watered every day with freshly produced PAW and the control treatment with tap water. A significant increase in G-POX activity for the PAW treatment was noticed in the young roots (Figure 1b), but the increase was not as big as in the corns; the difference between the control and PAW was an 80% increase (Figure 1b). On the other hand, no significant change in POX activity after 3 days of cultivation was noticed in the shoot.

The growth of primary seminal root and the shoot after the treatment with PAW improved in comparison with control plants (Figure 2). The increase in the PAW treatment was 13% in both roots and the shoot.

Besides POX activity and growth, root tissue development was also accelerated after the PAW treatment (Figure 3). The developmental stages of the cell wall in terms of its lignification or suberization of selected tissues, such as exo- and endodermis and xylem vessels, were compared in the roots exposed to PAW and the control roots. At a distance of 10% from the root apex, Casparian bands in endodermis and exodermis were detected in the roots treated with the PAW; cell wall lignification was also observed in the early metaxylems (Figure 3a).

On the contrary, in the control roots, the development of the cell walls in terms of their lignification was obviously decelerated (Figure 3b). These differences disappeared at the root base, where similar developmental stages of the exo- and endodermis and xylem elements were observed (Figure 3c,d). The findings were confirmed by staining with 4 MN detecting POX activity in situ in the cell walls, which was also associated with lignification (Figure 4). A blue color was present in the early metaxylems and endodermis of the PAW treatment (Figure 4a) and only in early metaxylems in the control roots (Figure 4b) indicating the lignification process of the cell walls.

To evaluate the potential of RONS in PAW in priming of the plants facing abiotic stress from arsenic (As^V+^), the pre-treatment in the paper rolls was broadened by the treatment named priming PAW (as defined in chapter 4.8). In this case, the corns were only imbibed in the PAW and then cultivated in paper rolls with tap water for three days. Contrary to this, in the rolls PAW treatment, the corns were imbibed in PAW and the paper rolls were watered every day with the freshly prepared PAW. The control was imbibed and cultivated in tap water. After this pre-cultivation, 3 day-old seedlings were grown in the hydroponics for another 10 days without and with As and were subsequently analyzed. Interestingly, cultivation with 150 μM As in the As treatment did not influence the growth of maize roots and the shoot (FW per one plant) negatively in comparison with the control (Figure 5a). Contrary to this, roots in the priming PAW, priming PAW As, and rolls PAW had higher fresh biomass (FW per one plant), probably associated with the water management, because the roots of these three treatments did not achieve a higher percentage of the dry biomass accumulation (% DW) than the control (Figure 5b). Shoots accumulated significantly more fresh weight (FW per one plant) than the control only in the priming PAW As and the rolls PAW treatments, but the accumulation of the dry weight (% DW) decreased in all treatments in comparison with the control. The length of the primary seminal root was affected mostly negatively in comparison with the control (Figure 6a,b). Only roots of the rolls PAW treatment achieved the control root length. The overall worst shoot habitus was, however, noticed in the As treatment; the leaves were bigger in the priming PAW As and the rolls PAW As treatment than in the As treatment (Figure 6a).

Changes in the root morphology were also confirmed by software analysis (Table 1). The observed characteristics of roots in all PAW and As treatments were different in comparison with the control. Arsenic, in all As treatments, caused a highly significant decrease in the number of root tips and the average diameter of the root, but increased the branching frequency in comparison with the control. The least root tips were found for the priming PAW As treatment and the tiniest roots (their average diameter) for the As and the rolls PAW As treatments. A comparison of the number of root tips between the PAW treatments and its corresponding As treatments showed the decreasing tendency in the case of priming, but an increase in the case of rolls. On the contrary, branching frequency increased due to the As treatment in the priming As and decreased in the rolls PAW As treatment.

Using two-way ANOVA analysis, we compared the significance of the two selected factors on the measured POX and CAT activities; the first one was a plant organ (roots versus the second leaf) and the second was a treatment type. After hydroponic cultivation, roots were identified as organs with significantly higher POX activity in comparison with the second leaf (Figure 7), and the priming PAW was the treatment with the highest POX activity followed by the As treatment and the control. The rolls PAW, the priming PAW As and the rolls As treatments had the lowest POX activity. One-way ANOVA of G-POX activity was performed and compared separately in the roots and in the second leaf. In the roots, the only significant increase in the POX activity was observed in the priming PAW treatment in comparison with the control. On the contrary, in the priming PAW As, the rolls PAW, and the rolls PAW As treatments, a significant decrease was achieved (Figure 7). In the second leaf, a significant increase in the POX activity was detected only in the As treatment in comparison with the control.

Contrary to POX activity, the two-way ANOVA with the same two selected factors did not show any difference between catalase (CAT) activities when comparing roots and the second leaf (Figure 8). When the factor of treatment type was assessed, the As, the rolls PAW As, and the priming PAW As treatments had significantly higher CAT activity than the priming treatment. The content of photosynthetic pigments Chl *a*, Chl *b*, and carotenoids was, in all treatments, affected negatively in comparison with the control. The only exception was, interestingly, the rolls PAW As treatment, where a significant increase in all tested pigments was noticed (Figure 9). The As treatment had the most noticeable decrease in all pigment concentrations. In the priming PAW, the priming PAW As, and the rolls PAW treatments, the concentration of Chl *a* and carotenoids was statistically the same and was decreased in comparison with the control.

Plants from the treatments without As (control, priming PAW, and rolls PAW) accumulated only a trace amount of As (data not shown), while all As treatments accumulated a significant amount of As, especially the roots, when compared to the first and the second leaves (Figure 10). In the roots, the highest concentration of As was found in the As treatment compared to the priming PAW As and the rolls PAW As treatments. Plants of the As and the priming PAW As treatments accumulated more As in the first leaf than in the second leaf. An opposite effect was found in the rolls PAW As treatment, where significantly more As was deposited into the second leaf. When calculating the ratios between As concentrations in the root and the selected leaves, we noticed 34.8, 21, and 31 between the root and the first leaf in the As, the priming PAW As, and the rolls PAW As treatments, respectively. This ratio was very different when considering the second leaf, where it was 80, 63, and only 6.6 in the As, the priming PAW As, and the rolls PAW As treatments, respectively. Another interesting fact arising from these results was the overall As accumulation in the roots and the selected leaves; the highest concentration was noticed in the As treatment.

After cluster analysis, groups with similar characteristics were grouped (Figure 11). To form the clusters, the procedure began with each observation in a separate group. Observations of the As and the priming PAW As treatments were the closest, forming a group along with the rolls PAW treatment. Interestingly, the rolls PAW As treatment and the control formed another group, although not so closely related according to the distance between them.

Another instructive result of this analysis is the separation of the priming PAW treatment, indicating a special position of this treatment in the observed experiments. The PCA analysis revealed a clustering of the plants from the rolls PAW, the As, and the priming PAW As treatments, while POX and CAT activities in the second leaf, CAT activity in the roots, and the % DW of the roots seem to play the most important roles as measured variables in these treatments. The rolls PAW As and the priming PAW treatments and the control were separated from each other, with different variables having the greatest impact. This analysis confirmed the correlation between some of the observed variables (Figure 12). A positive correlation was observed between the As concentration in the roots and in the first leaf (R^2^ = 0.89), between the As concentration in the second leaf and the carotenoids (R^2^ = 0.71), between the CAT activity in the second leaf and the roots (R^2^ = 0.90), between the Chl *a* and Chl *b* (R^2^ = 0.84), and between the Chl *a* or Ch *b* concentration and the carotenoids (R^2^ = 0.81 and 0.74, respectively). The only two significantly negative correlations were confirmed between the FW of the roots and CAT activity in the roots (R^2^ = −0.88) and between the % DW of the roots and POX activity in the roots (R^2^ = −0.67).

## 3. Discussion

Plasma activated water is a source of various reactive oxygen and nitrogen species (ROS and RNS) that can improve plant growth in stress conditions and partially replace the use of fertilizers. It is the priming effect of H_2_O_2_ that provokes plants to react faster and stronger to a potential stress, and NO_3_^−^ and NO_2_^−^ as a source of critically important microelement essential for building proteins and other macromolecules. Some authors describe the effects of PAW as alternatives to chemical biostimulators in very early embryo development, e.g., during seed germination (e.g., [42]), although the effect of PAW depends on several factors, such as plant species, plasma activated water activity (its chemical composition), and other experimental conditions [24]. Maize used in the present study has, in general, a high percentage of germination, and the difference between the PAW treatments and the control with respect to germination was non-significant (data not shown). However, what is shown here is that POX activity was strongly enhanced after just 6 h of corn imbibition in PAW in comparison with those imbibed in tap water (Figure 1a). Corona-Carrillo et al. [43] described the paradoxical role of POX in the developing maize embryo and claim that these enzymes can either produce or decompose ROS via their peroxidative and hydroxylic cycles, maintaining ROS and nutrition at optimum levels. Enhanced POX, an important antioxidant enzyme, avoids potential embryo damage and counteracts the stress occurring during germination or seeds storage. This can increase the embryo vigor and its capacity to establish seedlings [43]. Improved growth of seedlings after imbibition in PAW was confirmed in the present study (Figure 2), and significantly enhanced POX activity in the PAW treatment remained in the roots (Figure 1b). This is very important for plant survival in the subsequent development, because roots are most often the first contact with potential soil stressors [44,45]. The roots of seedlings treated with PAW developed faster (Figure 3), which corresponds to tissue lignification, which was delayed in the control (Figure 3a,b). The lignification of exo-, endodermis, and xylem vessels is one of the most important processes in the roots, because it controls apoplasmic ion flow and protects the central stele from entering the toxic elements in the xylem followed by shoot translocation [32,46,47,48,49,50]. On the other side, xylem lignification is an assumption of water transpiration that is essential for plant survival. In the 3 day-old seedlings, we detected in situ POX activity in the root cell walls (Figure 4) with elevated reactions in the endodermis of the PAW treatment (Figure 4b), which is associated with lignification. Peroxidases are key players in this process, polymerizing lignin precursors, monolignols [31]. Improved growth after the PAW treatment was also reported by [24] on lettuce (*Lactuca sativa*); however, changes depended on the harvest date and on the used PAW composition. A positive effect of PAW on the growth of another important crop, a wheat (*Triticum aestivum*), was documented by [42], which also confirmed a positive effect of PAW on photosynthetic pigment concentration but detected lowered activities of antioxidant enzymes.

Another objective of the present study was to document the effect of the PAW pre-treatment either at the level of 6 h corn imbibition (priming PAW treatment) or stimulation of seedlings during the 3 days of cultivation with PAW (rolls PAW treatment) followed by As stress during the other 10 days of hydroponic cultivation (Figure 5, Figure 6, Figure 7, Figure 8, Figure 9, Figure 10, Figure 11 and Figure 12). Thus, the tested plants in the priming PAW and the rolls PAW treatments were exposed to different doses of PAW, containing H_2_O_2_ and NO_x_^−^. Hydrogen peroxide has a special position among ROS. It has a dual role; at low concentrations it acts as a signal molecule triggering the antioxidant systems, but at high concentrations, it has destroying effects on cells [51]. Roots of the priming PAW and the rolls PAW treatments and shoot of the rolls PAW treatment accumulated more water than the controls (Figure 5a), but the % DW was not improved in comparison with the control (Figure 5b). Similarly, an acceleration in water uptake was documented by [12] on pea seeds exposed by atmospheric plasma. The length of the primary root of PAW treated plants was not greater than that in the control. However, plant root morphology was altered dramatically (Table 1). Due to the PAW treatment, roots had significantly less total length, less lateral roots, the least in the rolls PAW treatment, but the branching frequency increased in both PAW treatments. Additionally, the average diameter decreased in comparison with the control root. All these morphological changes are important characteristics and influence the accessibility and mobility of ions with different properties [52]. Bafoil et al. [53] tested the effects of PAW on the model plant *Arabidopsis thaliana* and confirmed a significant increase in various plant growth parameters.

Plants, with their sessile way of life, must face various stresses in their environment by activating the defense mechanisms. These reactions, unfortunately, often contribute to a decreased yield, a problem that can be solved by various approaches. One of them could be a use of PAW, which was reported to improve the tolerance against stress from low temperature and hypoxia during barley germination [54]. In the present study, As stress reaction was evaluated. Basic growth characteristics were negatively changed by As itself, which is a common phenomenon of this toxic metalloid [28,55] (Figure 5 and Figure 6, Table 1). Plants of the priming PAW As treatment accumulated more water than As-treated plants (Figure 5a), but no improvement of the FW was detected in the rolls PAW treatment. Similarly, the % DW was the same when comparing the As and other treatments (Figure 5b). In general, in several cases we found different plant response to the As treatment when the priming PAW As and the rolls PAW As treatments were compared to each other, to the As treatment or to the control. This phenomenon could be explained by various doses of H_2_O_2_ and NO_x_^−^ given to corns or plants during pre-treatment, where the priming treatments received PAW only during imbibition, and the rolls treatments also received it during the following 3 days of cultivation. The divergent reactions were finally confirmed, also using multivariate statistical analysis (Figure 11 and Figure 12), where treatments separated from each other in a specific way. Maintaining cell homeostasis during a stress reaction is a key in enabling plants to survive any sub-optimal conditions. To keep ROS content at the non-toxic level, the involvement of antioxidant enzymes is essential; however, stimulation of their activity is not always obvious [28,32,56,57]. A common phenomenon of significantly higher POX activity in the roots than in the leaves was also confirmed in the present study (Figure 7). Interestingly, when results were analyzed by two-way ANOVA, and factor treatment was evaluated, the rolls PAW As, the priming PAW As, and the rolls PAW had the lowest POX activities. The control and the As treatment were in the second group and the highest activity had plants of the priming PAW treatment. POX activity showed that roots reacted to the As treatment non-significantly; however, a significant increase was detected in the second leaf in comparison with the control (Figure 7). A significant increase in POX activity in the roots was detected only in the priming PAW treated roots, while all other roots had decreased POX activity. The production of antioxidants in the plants challenged by As is a common phenomenon [58]. However, a divergent reaction of POX activity and pattern in POX expression was documented, caused by different As doses in tobacco (*Nicotiana benthamiana*) plants [28]. It is clear that plants reacted with elevated POX activities on moderate As stress, but the reaction was different for low and high As doses. We suppose that, in the present study, a similar phenomenon occurred, i.e., plants experienced stress not only from As, but also from PAW containing H_2_O_2_. Contrary to POX, when we documented the activity of the collaborating antioxidant enzyme CAT [59], no difference was noticed when roots and the second leaf were compared (Figure 8). The obvious cooperation between the two enzymes’ decreasing H_2_O_2_ levels was detected in the priming PAW treatment. It was the only case of decreased CAT activity in comparison with other treatments and, we suppose, the major role of POX reducing H_2_O_2_ in this treatment. Changes in CAT activity after the PAW treatment was also observed in other plant species [53,60]; however, Gierczik et al. [54] documented the time-dependent increase or decrease. It is obvious that the relationship between the CAT (de)activating and H_2_O_2_ content is time dependent, and another regulator, the ascorbate-glutathione cycle, could also be involved [61].

Plants also possess non-enzymatic antioxidants, such as pigments–carotenoids, preventing oxidative burst [62]. The rolls PAW As was the only treatment where we detected a significant increase in these pigments (Figure 9), and at the same time, the second leaf of this treatment had the highest chlorophyll contents; in this case, they were probably the best protected by carotenoids. These three pigments were closely correlated (Figure 12) in the present study, and the relationship between carotenoids and As acting was also confirmed by the detection of a significant negative correlation between As content in the second leaf and carotenoids (Figure 12). The alleviation of a salt-induced damage by PAW through its effect on carotenoids content has also been described [54]. Authors also proved a slight increase in the glutathione metabolism-related genes due to the PAW treatment, which indicates changes triggered by signal molecules produced in PAW at the DNA expression level.

The accumulation of As in the aboveground edible plant part is always dangerous due to the carcinogenic character of this metalloid [63]. The accumulation of As in the present study was not affected by the PAW treatments (Figure 10). Unfortunately, higher amounts were accumulated in both tested leaves of the priming PAW As and the rolls PAW As treatments. Changes in the As uptake, radial transport, and its translocation are a combination of the altered root morphology, anatomy, water transport management, and As transporters capacity. To explain this phenomenon completely, additional research in this field is necessary in the future. Here, we have only partially analyzed the formation of apoplasmic barriers in the roots due to PAW treatments, but we did not get enough data to draw further conclusions.

## 4. Materials and Methods

### 4.1. Plant Material

Corns of maize (*Zea mays* L.) (hybrid Bielik) used in the experiments were obtained from Sempol spol. s.r.o., Bratislava, Slovakia. Hybrid Bielik was selected after screening of several maize hybrids exposed to PAW. The corns were stored in fridge at 8 °C in the dark. All treatments were repeated at least three times, independently.

### 4.2. Production of Plasma Activated Water (PAW)

The plasma activated water was generated by a transient spark discharge operated in atmospheric air in a contact with tap water. The plasma reactor was of a point-to-plane geometry and consisted of a needle used a high voltage electrode placed above an inclined grounded electrode embedded in a polytetrafluoroethylene gutter. Tap water was driven down the gutter, repetitively circulated, and exposed to the transient spark discharge for a given time. The transient spark (TS) discharge is a repetitive streamer-to-spark transition discharge. It is a DC-driven, self-pulsing discharge typical of current pulses of high amplitude (order of several tens of amps), very short duration (10–100 ns), and frequency of orders of several kHz. The details on physical properties of TS [64,65] as well as on a system for plasma activated water generation [42,66,67] have been previously published. In the presented study, the transient spark discharge was operated at the applied voltage 11–13 kV, amplitude, and frequency of current pulses ~3 A and 1.5–3 kHz, respectively. The water flow was set to 15 mL min^−1^. The amount of tap water exposed to plasma varied; however the plasma exposure/activation time was set to 1 mL min^−1^, i.e., 20 mL of water was exposed to plasma for 20 min, etc. The experiments were performed at ambient air temperature ~22 ± 2 °C. The temperature of water was maintained by an ice bath to avoid unwanted heating of the produced PAW caused by the plasma exposure. The operating conditions were alike those we used in past experiments [42,67] and which turned out to be optimal for the stimulation of plants by PAW. Plasma activation of tap water did not affect its pH (7.5) due to natural hydrocarbon buffer system; however, it resulted in the formation of various RONS in water whose concentrations were evaluated.

### 4.3. Measurement of Hydrogen Peroxide (H_2_O_2_), Nitrate (NO_2_^−^) and Nitrite (NO_3_^−^)

The concentrations of RONS, namely hydrogen peroxide (H_2_O_2_), nitrate (NO_2_^−^), and nitrite (NO_3_^−^), in PAW were measured by UV–Vis absorption spectrometer UV-1900 (Shimadzu, Japan). Hydrogen peroxide (H_2_O_2_) is mostly produced in gas phase by a recombination of OH radicals and subsequently dissolves in water. Its analysis is based on its reaction with titanyl ions (Ti^4+^) of titanium oxysulfate (TiOSO_4_) [68]. The reaction results in the formation of a yellow-colored product of pertitanic acid (H_2_TiO_4_) with a maximum absorbance peak at 407 nm proportional to H_2_O_2_ concentration. The reaction is specific to H_2_O_2_ and does not interfere with other compounds. Prior to H_2_O_2_ analysis, the PAW was fixed with sodium azide (NaN_3_) to eliminate its eventual decomposition by a mutual reaction with NO_2_^−^. NaN_3_ reduces NO_2_^−^ to molecular N_2_ and does not interfere with the H_2_TiO4. The nitrites (NO_2_^−^) and nitrates (NO_3_^−^) in PAW are mainly formed by the dissolution of gaseous HNO_2_ and HNO_3_ formed in a gas phase. They may also form by NO_2_ dissolution, however, which is less efficient than the dissolution of the corresponding acids. The analysis of NO_2_^−^ and NO_3_^−^ was performed using commercially available kits based on reaction with sulfanilamide and N-1-naphthylethylenediamine, the so-called Griess reagents [69]. The Griess reagents react with NO_2_^−^ to form a pink-colored azo-product with a maximum absorbance peak at 540 nm. To measure NO_3_^−^ concentration, it must first be enzymatically reduced to NO_2_^−^ and then analyzed by the same method as that of NO_2_^−^. The method is easy to perform and is approved as being precise for NO_2_^−^ and NO_3_^−^ measurements in PAW. 

### 4.4. Maize Corns and Seedlings Treated with PAW

Dry maize corns were imbibed in tap water or plasma activated water (PAW) for 6 h at room temperature. Several corns (two to three) were randomly chosen for guaiacol peroxidase (G-POX, POX) activity measurement. The rest of imbibed grains (30 for each variant) were wrapped in wet sterile filter paper and cultivated. In this part, two treatments were established: control (corns imbibed in tap water, and then in paper rolls watered daily with tap water) and PAW treatment (corns imbibed in PAW, and then in paper rolls watered daily with freshly prepared PAW). Seedlings were cultivated for three days in the dark under controlled physical conditions in an incubator at the temperature 24 ± 2 °C and 60% relative humidity. At the end of the cultivation, young roots and shoot were measured and used for G-POX activity measurement, lignification of the root tissues, and in situ POX activity in roots.

### 4.5. G-POX Activity in Corns and Seedlings

After three days of cultivation, samples of the roots and the shoot were randomly chosen from at least four plants (~1.5 g), were ground with a mortar and pestle in liquid nitrogen, and suspended in 50 mM Na-phosphate protein extraction buffer with 1 mM EDTA (ethylenediaminetetraacetic acid), pH 7.8. After 15 min centrifugation (12,000× *g*) at 4 °C, the supernatant was used for spectrophotometrical determination of total soluble protein concentration at 595 nm, according to [70]. Protein content was calculated as the total number of proteins per gram of fresh matter from the calibration curve with bovine serum albumin (BSA) as the protein standard. Guaiacol peroxidase (G-POX, POX, E.C.1.11.1.7) was measured according to standardized assays, with a minimum of three measurements and three technical replications per each sample. The activity of G-POX was established according to [71] and measured spectrophotometrically at 440 nm. The G-POX activity was expressed in nM of tetraguaiacol min^−1^ mg^−1^ multiplied by the molar extinction coefficient of tetraguaiacol 26.6, as follows:$$specific\;G-POX\;activity=\frac{∆\;A{\;\min}^{-1}\times 1000}{26.6\;\frac{protein\;content\;in\;sample\;\left(\mu g\right)}{volume\;of\;extraction\;solution\;\left(\mathrm{mL}\right)}}\;(\mathrm{nM})$$

### 4.6. Lignification of the Root Tissues

In the three-day-old seedlings, influence of the PAW treatment was observed on the lignin deposition on the free hand sections of the roots at a distance of 10% from the root apex and at the root base. This approach allows analysis and comparison of the same developmental stages of the root, irrespective of the root length differing between treatments. The sections were stained with fluoroglucinol–HCl for visualization of lignin deposition. Fluorescence of the lignin deposits was observed by an Axioskop 2 plus microscope (Carl Zeiss, Jena, Germany), equipped with excitation filter TBP 400 + 495 + 570 nm, chromatic beam splitter TFT 410 + 505 + 585, and emission filter TBP 460 + 530 + 610 nm, documented by an Olympus DP 72 camera system and analyzed with Lucia imaging software (Lim, Prague, Czech Republic). Four roots per treatment were analyzed in each experimental run.

### 4.7. In Situ POX Activity in Roots

To document the activity of cell wall peroxidases in situ, the hand cross root sections (~0.5 mm thick) 10% from the root apex were observed. The sections were incubated in 100 mM Na-acetate buffer (pH 5.2) with 5 mM 4-methoxy-1-naphthol (4-MN) in 96% ethanol for 15 min at 30 °C [72,73], observed in the bright field and documented as described in chapter 4.6. Four roots per treatment were analyzed in each experimental run.

### 4.8. Maize Seedlings Treated with Arsenic (As) after PAW Pre-Treatment

To document a potential of PAW in priming maize corns and seedlings subsequently exposed to stress from arsenic (As), the previous two treatments in the paper rolls (chapter 4.4) were broadened to three treatments: the control (corns imbibed in tap water, seedlings cultivated for three days in paper rolls with daily fresh tap water); priming PAW (corns imbibed in PAW, seedlings cultivated as control); and rolls PAW (corns imbibed in PAW, seedlings cultivated for three days in paper rolls with daily freshly prepared PAW—in chapter 2.4 called PAW treatment). Ten randomly selected 3 day-old seedlings from each treatment were transferred into 3 L containers and cultivated for other 10 days as hydroponic cultures in 0.5 Hoagland nutrient solution (pH 5.8). The concentration of arsenic (150 μM As) was established based on a previous concentration screening. It was added in the form of As^5+^ (Na_2_HAsO_4_.7H_2_O) and, finally, six treatments in total were established: the control (0.5 Hoagland solution, seedlings from control treatment), As (150 μM As, seedlings from control treatment), priming PAW (seedlings from priming PAW treatment), priming PAW As (150 μM As, seedlings from priming PAW treatment), rolls PAW (seedlings from rolls PAW treatment), and rolls PAW As (150 μM As, seedlings from rolls PAW treatment). Hoagland nutrient solution without or with As was changed every 3 days. At the end of the cultivation, plants were evaluated in terms of their growth parameters—fresh weight (FW), % of the dry weight (DW) expressed as the ratio of FW and DW multiplied by 100, and length of the primary seminal root—and other biochemical characteristics were detected. Macrophotography images of plants in individual treatments were taken using a camera Nikon D90 with an AF-S Micro Nikkor 60 mm lens system. Roots of all treatments were also scanned in high quality, and the images were analyzed by RhizoVision Explorer to obtain other root characteristics, such as total root length, diameter, branching frequency, and number of root tips.

### 4.9. POX and CAT Activity in Maize Seedlings

The below- and above-ground plant parts of the 10 day-old seedlings were detached, and the roots were thoroughly washed three times in distilled water. Roots and the 2nd leaf of at least 4 randomly selected plants were used for assays of POX activity in the same way as described in chapter 4.5. Extract of proteins from roots and the 2nd leaf were used also for catalase (CAT) detection [74]. Its activity was calculated after spectrophotometrical measurement at 240 nm, based on the decomposition rate of H_2_O_2_ in time [73].
$$specific\;catalase\;activity=\frac{∆\;A{\;\min}^{-1}\times 1000}{39.1\;\frac{protein\;content\;in\;sample\;\left(\mu g\right)}{volume\;of\;extraction\;solution\;\left(\mathrm{mL}\right)}}$$

### 4.10. Evaluation of Chlorophylls and Carotenoids Concentration in Leaves

The 2nd leaf of every treatment was used for the determination of photosynthetic pigments concentration; leaves were randomly selected from at least four plants. Chlorophyll a, b (Chl *a* and Chl *b*), and carotenoids were extracted with the cooled mortar and pestle on the 10th day of cultivation (ca 500 mg of FW) of each treatment with cooled 80% acetone (10–15 mL) with 200 mg of MgCO_3_ mixed with a little sea sand to prevent phaeophytin formation. The pigment concentrations were determined spectrophotometrically (Jenway 6400, London, UK) as follows: Chl *a* at 663.2 nm, Chl *b* at 646.8 nm, and carotenoids at 470 nm. The concentrations were calculated after [74] and expressed as mg of pigment per 1 g of plant material fresh weight.
Concentration of Chl *a* = ((12.25 A663.2 − 2.79 A646.8) × y)/60 (mg g^−1^ FW)
Concentration of Chl *b* = ((21.50 A646.8 − 5.10 A663.2)) × y)/60 (mg g^−1^ FW)
Concentration of carotenoids = ((1000 × y* A470 − 1.82 Chl *a* mg L^−1^ − 85.02 Chl *b* mg L^−1^)/198))/60 (mg L^−1^)
where
y = (the volume of acetone used for extraction × 0.06)/FW of material (g),
Chl *a* mg L^−1^ = (12.25 A663.2 − 2.79 A646.8) × y
Chl *b* mg L^−1^ = (21.50 A646.8 − 5.10 A663.2) × y

### 4.11. Determination of As Concentration in Leaves and Roots

Concentrations of As in the 1st oldest leaf, in the 2nd leaf, and in the roots were determined in each As treatment by atomic absorption spectrometry (AAS). Dry plant samples taken randomly from the plants were dried at 70 °C until constant weight. At least 200 mg of DW was used for As determination. Samples were dissolved in concentrated HNO_3_. After heating at 160 °C for 3 h, concentrated HF was added. Thereafter the samples were dried, and a mixture of concentrated HNO_3_ and H_3_BO_3_ was added. The control and the PAW treatments were also checked for As content—plants accumulated only trash amounts of this metalloid.

### 4.12. The Statistical Analysis

All results were evaluated using PC software Excel with XLSTAT (Microsoft Office 365, Redmond, Washington, DC, USA) and statistic software Statgraphics Centurion XVI. Analysis of variance (ANOVA, One-Way, and Multifactorial) with LSD test, standardized clustering method—nearest neighbor (distance metric: squared Euclidean), and principal component analysis (PCA) with the level of significance *p* < 0.05 (significant) were performed. All experiments were conducted independently three times and results were expressed as means of four replicates ± standard deviations (SD) in the figures.

## 5. Conclusions

The effect of PAW generated by a transient spark discharge (TS) operated in ambient air on maize corns and young seedlings was investigated and confronted with the effect of As stress in several different treatments. PAW treatment provoked the enhancement of POX activity immediately after the corn imbibition, which points to the influence of highly active molecules (RONS) within this solution. PAW itself improved the growth parameters only in the young seedlings; however, in continuous hydroponic cultivation, plants also achieved positive changes in the dry biomass accumulation. The primary root morphology also changed due to PAW treatments, which significantly influence the plant nutrients uptake. Together with the enhanced photosynthetic pigments, it can be concluded that treatments with PAW contribute to better survival of the plant. We also found a different pattern of plant response to the subsequent As treatment. Plants from different PAW treatments reacted to the As stress by elevating their antioxidant capacities; depending on PAW pre-treatment, antioxidant enzymes, POX and CAT, or non-enzymatic molecules, carotenoids were elevated, reflecting the active defense system activated due to PAW pre-treatment. The study confirmed that the use of PAW in the pre-treatment (priming) of maize corns or young plants may improve their tolerance against As stress, which can be utilized when plants are grown in toxic elements contaminating soils.

## Figures and Tables

**Figure 1 plants-10-01899-f001:**
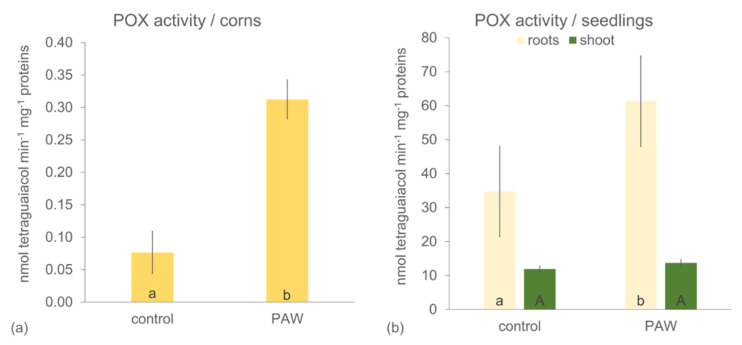
Activity of guaiacol peroxidase (G-POX, POX) in maize corns (hybrid Bielik) after 6 h imbibition (**a**) and in the roots and the shoot after 3 days of cultivation in the paper rolls; and (**b**) in the control (tap water) or treated with PAW. Values are means of four replicates ± SD. Different letters denote a significant difference between the treatments.

**Figure 2 plants-10-01899-f002:**
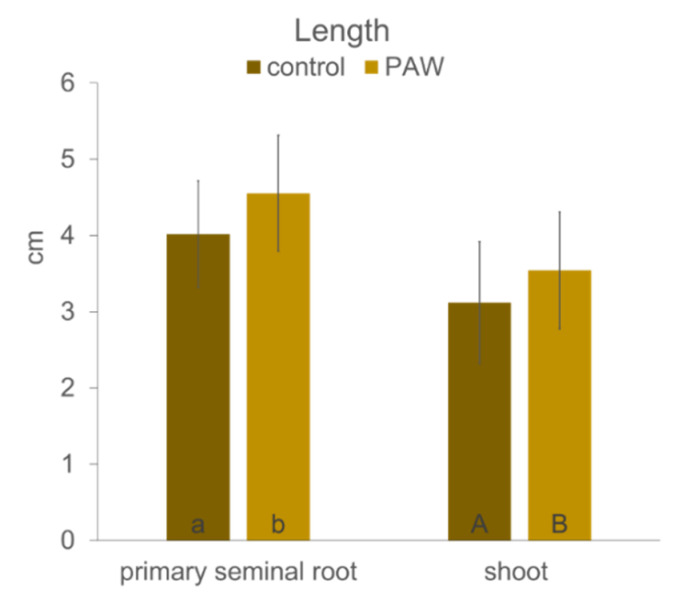
Growth of the primary seminal root and the shoot of maize (hybrid Bielik) after 3 days of cultivation in the paper rolls as control or treated with PAW. Values are means of four replicates ± SD. Different letters denote a significant difference between the treatments.

**Figure 3 plants-10-01899-f003:**
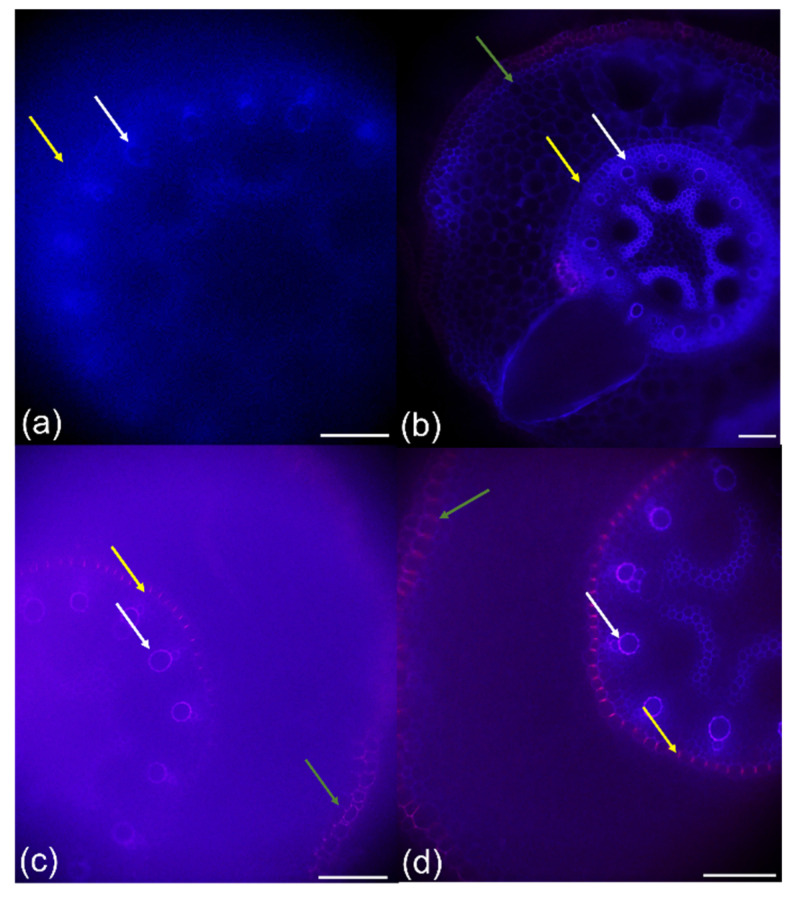
Lignification of the maize roots (hybrid Bielik) after 3 days of cultivation in the paper rolls as control (**a**,**c**) or treated with PAW (**b**,**d**). The hand cross sections were 10% from the root apex (**a**,**b**) or on the root base (**c**,**d**) and stained with phluoroglucinol-HCl. The arrows point at the exodermis (green), endodermis (yellow), and early metaxylem (white). Scale bars = 100 μm.

**Figure 4 plants-10-01899-f004:**
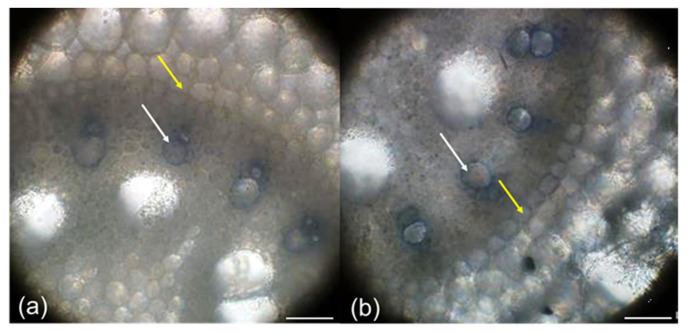
In situ POX activity in the maize roots (hybrid Bielik) after 3 days of cultivation in the paper rolls as control (**a**) or treated with PAW (**b**). The hand cross sections were 10% from the root apex and stained with 4 MN. The arrows point at the endodermis (yellow) and early metaxylem (white). Scale bars 100 μm.

**Figure 5 plants-10-01899-f005:**
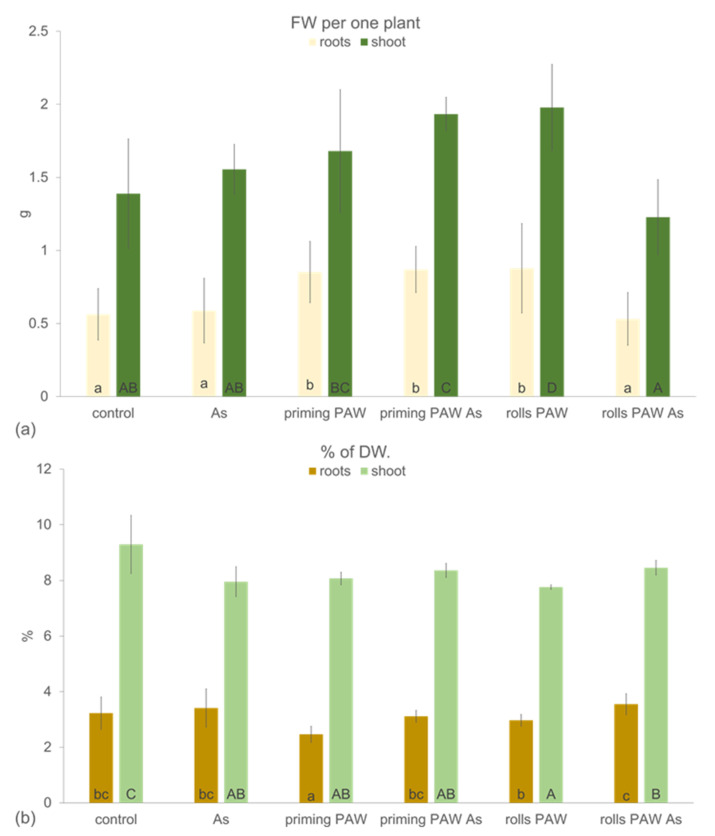
Growth parameters of the maize plants (hybrid Bielik) after 10 days of hydroponic cultivation as control or pre-treated with PAW (priming PAW and rolls PAW treatments) without and with As; the fresh weight per one plant (**a**) and the % of the dry weight (**b**). Values are means of four replicates ± SD. Different letters denote a significant difference between the treatments.

**Figure 6 plants-10-01899-f006:**
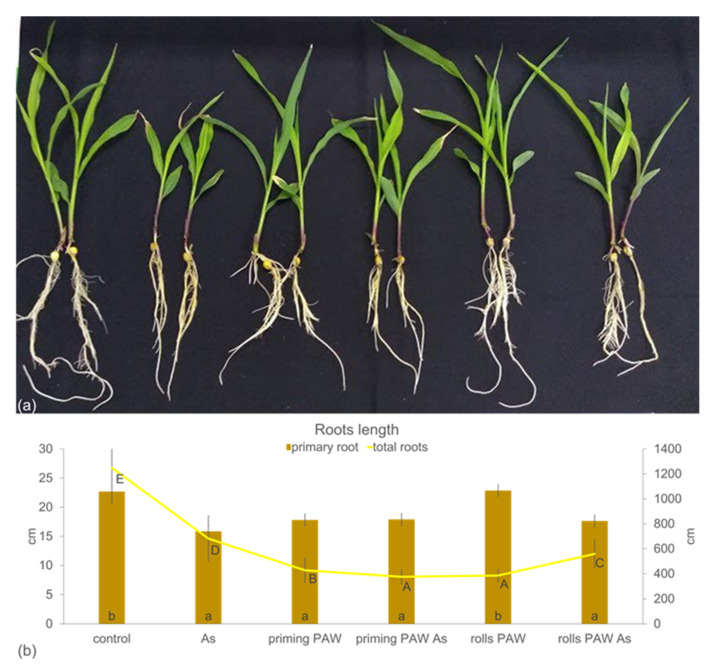
Habitus (**a**) of the maize plants (hybrid Bielik) and roots growth parameters (**b**) after 10 days of hydroponic cultivation as control or pre-treated with PAW (priming PAW and rolls PAW treatments) without and with As. Values (columns for primary root, line for total roots) are means of four replicates ± SD. Different letters denote a significant difference between the treatments.

**Figure 7 plants-10-01899-f007:**
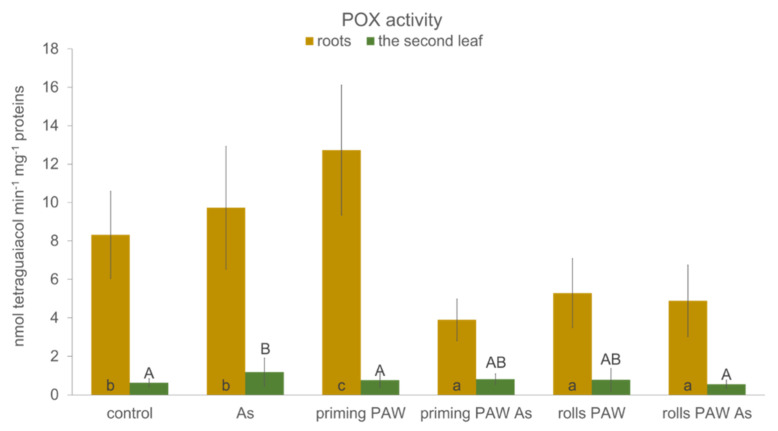
Activity of guaiacol peroxidase (G-POX, POX) of the maize roots and the second leaf (hybrid Bielik) after 10 days of hydroponic cultivation as control or pre-treated with PAW (priming PAW and rolls PAW treatments) without and with As. Values are means of four replicates ± SD. Different letters denote a significant difference between the treatments.

**Figure 8 plants-10-01899-f008:**
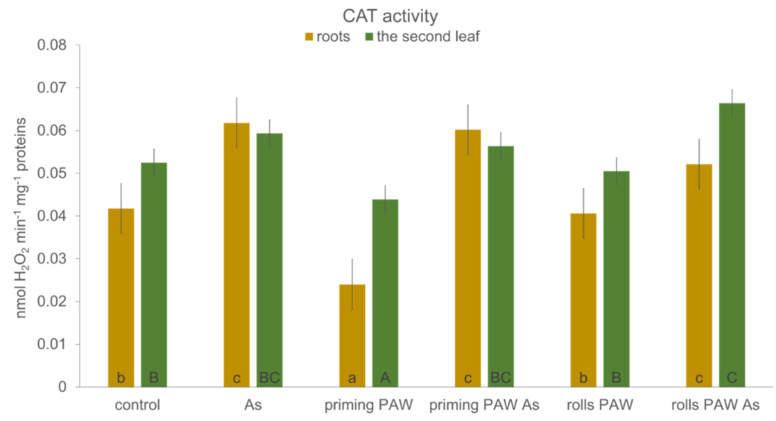
The catalase (CAT) activity of the maize roots and the second leaf (hybrid Bielik) after 10 days of hydroponic cultivation as control or pre-treated with PAW (priming PAW and rolls PAW treatments) without and with As. Values are means of four replicates ± SD. Different letters denote a significant difference between the treatments.

**Figure 9 plants-10-01899-f009:**
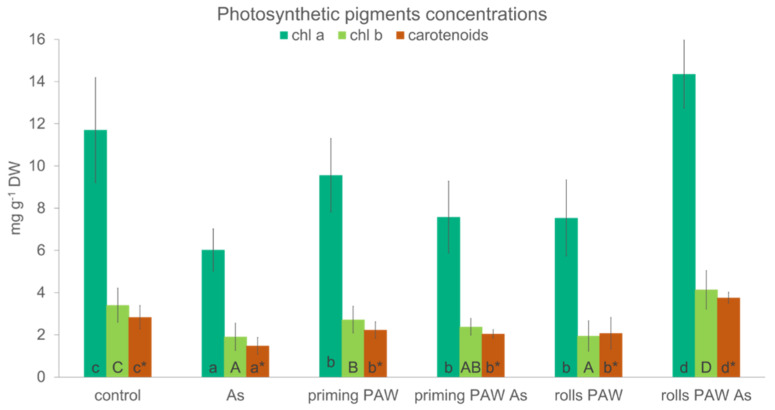
The photosynthetic pigments concentration per dry weight of the second leaf (hybrid Bielik) after 10 days of hydroponic cultivation as control or pre-treated with PAW (priming PAW and rolls PAW treatments) without and with As. Values are means of four replicates ± SD. Different letters denote a significant difference between the treatments.

**Figure 10 plants-10-01899-f010:**
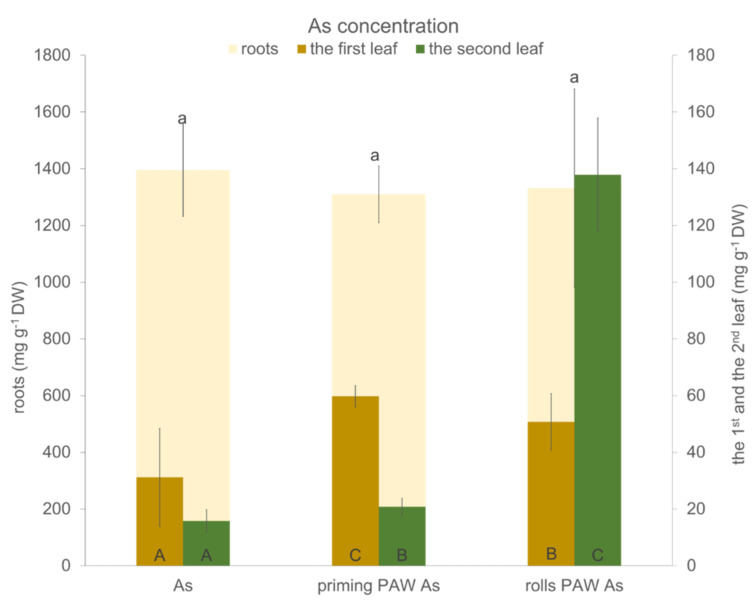
The As concentration per dry weight of the roots, the first leaf, and the second leaf (hybrid Bielik) after 10 days of hydroponic cultivation as As treatment or As treatments pre-treated with PAW (priming PAW As and rolls PAW As treatments).

**Figure 11 plants-10-01899-f011:**
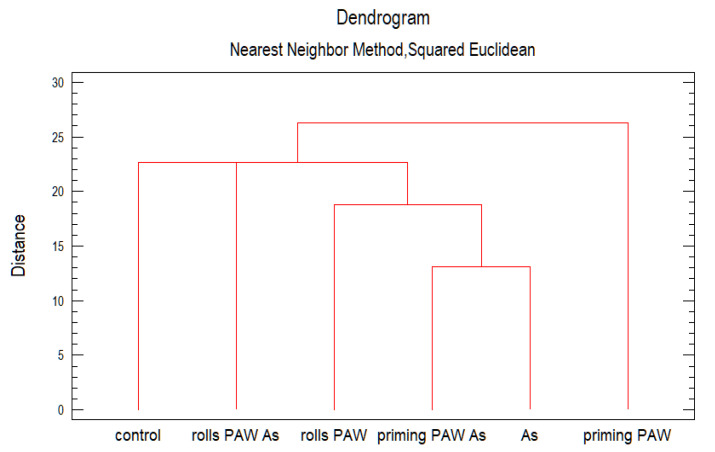
Cluster analysis. Clusters are groups of observations with similar characteristics according to the treatments.

**Figure 12 plants-10-01899-f012:**
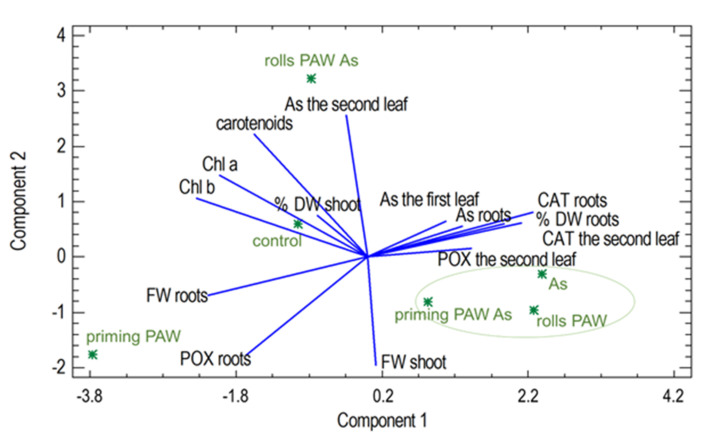
Biplot of the principal component analysis (PCA). The first two components extracted from the analysis together account for 61.2% of the variability in the original data.

**Table 1 plants-10-01899-t001:** Maize root (hybrid Bielik) morphological characteristics after 10 days of hydroponic cultivation as control or pre-treated with PAW (priming PAW and rolls PAW treatments) without and with As. Values are means of four replicates ± SD. Different letters denote a significant difference between the treatments.

Table	Number of Root Tips	Branching Frequency per mm	Average Diameter (mm)
control	785 ± 12 e	0.65 ± 0.02 a	0.76 ± 0.04 d
As	447 ± 25 d	0.87 ± 0.11 d	0.49 ± 0.05 a
priming PAW	191 ± 13 b	0.71 ± 0.05 b	0.6 ± 0.01 c
priming PAW As	146 ± 18 a	0.87 ± 0.08 d	0.53 ± 0.08 b
rolls PAW	185 ± 9 b	0.86 ± 0.05 d	0.51 ± 0.07 b
rolls PAW As	264 ± 14 c	0.81 ± 0.06 c	0.46 ± 0.1 a

## Data Availability

Not applicable.

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
