# Peer review of "The Effect of Plasma Activated Water on Maize (Zea mays L.) under Arsenic Stress"

_plants, 2021, doi:10.3390/plants10091899_

Round 1

Reviewer 1 Report

Arsenic is not essential element for plant growth, but it can accumulate in plants to toxic levels. As a result, it can enter the food chain and pose health risk to humans. It is commonly believed that PAW (plasma-activated water) possess outstanding biological activities in agricultural applications. PAW is utilized to increase the rate of germination of seeds and subsequent growth of seedlings and plants, inactivate plant-related pathogenic organisms and cure fungus-infected plants. In spite of the numerous positive effects of PAW on agricultural activation described in the literature, PAW interaction with biological objects including plant seeds, living cells or microorganisms remains not well understood. The submitted manuscript focuses on the elucidation of PAW interaction with the plant defensive systems and the estimation of PAW potential in priming of seedlings subsequently exposed to stress from As.  The presented results indicate that the As treatment, the PAW treatment and their combination influence growth parameters of the maize plants as well as POX and CAT activities in  a different pattern.

Specific question and comments:

Captions for Figures “Different letters denote a significant difference between the treatments.”

Where is this letter code explained?

Line 169-170 “Interestingly, cultivation with 150 µM As in the As treatment did not influence the growth of maize roots and the shoot (FW per one plant) negatively in comparison with the control (Figure 5 a). “

Why was 150 µM As used in the experiments if it had no effect on the growth of maize roots and the shoot ? Were other concentrations of As tested?

Line 256-258

“Plants of the As, and the priming PAW As treatments accumulated more As in the 1 st leaf than in the 2 nd leaf. An opposite effect was found in the rolls PAW As treatment, where significantly more As was deposited into the 2 nd leaf.”

Could it be concluded from Figure 10 that the priming PAW  treatment and the rolls PAW  treatment facilitate As transport form roots to aboveground parts of plants? Could this transport contribute to As storage within the plant’s edible parts?

“A promising way how to get plants with induced resistance against stress is pre-treatment (priming) of seeds or plants by exposure to a chemical compound acting as a stressor”

In the context of the results presented in Figure 10, is the priming PAW As treatment or the rolls PAW  As treatment a suitable approach to obtaining crops resistant to heavy metals?

Line 414-416

“Unfortunately, higher amounts were accumulated in both tested leaves of the priming PAW As and the rolls PAW As treatments. To explain this, additional research in this field is necessary in the future.”

Could it be elucidated, at least to some extent, by the following observation “Shoots accumulated significantly more fresh weight (FW per one plant) than the control in the priming PAW As …, but the accumulation of the dry weight (% DW) decreased in all treatments in comparison with the control” (Line 174-177)? This result may suggest that water transport is accelerated by PAW, which in turn can improve the mobility of arsenic ions within the shoot.

Author Response

 Dear reviewer, we would like to thank you for evaluation of the manuscript and all your valuable comments. We hope that our answers will be satisfying and explaining all the questions given in your review. We have tried to incorporate them also into the text of the re-submitted manuscript where they are marked in red.

Thank you very much.

Sincerely yours Zuzana Lukacova, corresponding author

Specific question and comments:

Captions for Figures “Different letters denote a significant difference between the treatments.”

Where is this letter code explained?

Thank you for this comment. Using different letters to denote the significant difference is a standard way how to show the results from the post-hoc tests. The treatments/mean values are statistically different when they are marked by different letter; in the case when there are more letters for one mean, there must not be any letter equal.

Line 169-170 “Interestingly, cultivation with 150 µM As in the As treatment did not influence the growth of maize roots and the shoot (FW per one plant) negatively in comparison with the control (Figure 5 a). “ Why was 150 µM As used in the experiments if it had no effect on the growth of maize roots and the shoot ? Were other concentrations of As tested?

Thank you for this comment. Toxic non-essential elements like arsenic together with heavy metals and their effects on the plants are one of the most studied topics in our working group. First of all we have tested several maize hybrids and their germination after PAW treatment. This selection is important because there are not only interspecific, but also different intraspecific reactions. The concentration of arsenic As used in the present study (150 mM) was also chosen after pre-screening. We needed to use a concentration that will somehow decrease the plant production parameters, but at the same time we needed the plants to survive. Moreover, this As content has been used also by our collaborators, so we wanted to stay consistent with them to be able to compare our results (these works are still in progress and are not published yet). It is still surprising, how different hybrids react differently. For instance, we found that some of them react on 150 mM As very negatively, i.e., after several days their growth almost completely stopped, and plants became necrotic. From that point of view, selecting Bielik hybrid and this concentration was optimal with respect to our cultivation period and methods. Another important point that we wanted to express was that only the fresh weights (FW) were not negatively influenced by As. Contrary to this, the % of the dry weight was negatively affected in the shoot what is much better indicator of the growth retardation, because the FW can reflect only the water accumulation, not the real growth. The negative effect of 150 mM As was also confirmed in the other characteristics such as total root lengths and the primary root length and, with no doubts, on the photosynthetic pigments concentration – especially chlorophyll a is very sensitive and is easily destroyed under stress.

Line 256-258

“Plants of the As, and the priming PAW As treatments accumulated more As in the 1 st leaf than in the 2 nd leaf. An opposite effect was found in the rolls PAW As treatment, where significantly more As was deposited into the 2 nd leaf.”

Could it be concluded from Figure 10 that the priming PAW  treatment and the rolls PAW  treatment facilitate As transport form roots to aboveground parts of plants? Could this transport contribute to As storage within the plant’s edible parts?

Thank you for this comment. Yes, from the results it seems that the As translocation was facilitated. However, this phenomenon has several more results: first of all, the overall As content in the roots and also in the shoot is lower in the PAW treated plants, because they have shorter roots and accumulate fewer dry weights in the aboveground plant parts. If this contributes to the translocation into the grain is questionable – we have tested very young plants so we cannot confirm this. On the other side, plants very often use the strategy of the toxic element accumulation in the oldest leaves that are the first being necrotised to avoid speeding it. Another phenomenon that we have observed is a premature root development, especially in term of lignification and apoplasmic barriers formation what can also contribute to the translocation restriction. This was confirmed in very young plants (3-day cultivation), but for 10-day hydroponic cultivation the research has not been completed yet. The root anatomy influenced by PAW deserves more attention and therefore will be investigated in near future.  

“A promising way how to get plants with induced resistance against stress is pre-treatment (priming) of seeds or plants by exposure to a chemical compound acting as a stressor”

In the context of the results presented in Figure 10, is the priming PAW As treatment or the rolls PAW  As treatment a suitable approach to obtaining crops resistant to heavy metals?

Thank you for this comment. As already mentioned, this research is a pioneer study dealing with As in a combination with PAW. The results have shown positive changes in the chlorophyll concentrations and in the antioxidant enzymes activities. If this contributes to really resistant crop we cannot confirm only based on the obtained results. Prolonged cultivation and testing a wider range of maize hybrids is necessary to see if it is a common phenomenon.    

Line 414-416

“Unfortunately, higher amounts were accumulated in both tested leaves of the priming PAW As and the rolls PAW As treatments. To explain this, additional research in this field is necessary in the future.”

Could it be elucidated, at least to some extent, by the following observation “Shoots accumulated significantly more fresh weight (FW per one plant) than the control in the priming PAW As …, but the accumulation of the dry weight (% DW) decreased in all treatments in comparison with the control” (Line 174-177)? This result may suggest that water transport is accelerated by PAW, which in turn can improve the mobility of arsenic ions within the shoot.

Thank you for this comment. The water accumulation in the shoot can be either due to acceleration of the water transport, or due to the decreased transpiration. However, we cannot distinguish and confirm it only based on our results. Maybe a research on the stomata cells is needed in the future. I cannot be sure if the increased translocation of arsenic into the shoot reflects only the water management of a plant. However, I believe it is a combination of this phenomenon, apoplasmic transport restriction in the radial root transport, root morphology changes (which we also confirmed) and arsenic transporters are also involved.

Reviewer 2 Report

Article reports the effect of plasma activated water (PAW) generated by a transient spark discharge on maize corns and young seedlings. In different treatments, the As stress was also investigated. Authors describe the plant response to the As treatment, the PAW treatment and their combination. Several treatments formed the most similar group as proven by cluster analysis. The As treatment provoked the increase of peroxidases and catalases in some leafs and the roots while the priming PAW treated roots had the highest activity.

As the main results, I consider:

  • PAW itself improved the growth parameters only in the young seedlings, however, in the continuous hydroponic cultivation, plants did not achieve significant increase in the biomass production.
  • The plants treated with the PAW alone or in a combination with As always had significantly higher concentration of photosynthetic pigments at the present higher As leaves accumulation.
  • It points at the protective mechanisms provoked by the PAW treatment.

The article is well written, not easy to understand, due to many of presented data, however, I don't think it may be presented in better form. I recommend its publication after following minor revision:

In the results, you present the concentration of NO2 and NO3 approx. 1 mM what correspond to pH approx. 3. However, you declare the pH as 7.5 (line 107), indicating the alkalic solution. Please explain this.

Author Response

We would like to thank you for your appreciation and valuable opinion on the submitted manuscript and a comment. In this short letter, we try to address it (original comment is in blue color). Based on our response, we also modified the text in the manuscript (track changes). We hope you will find changes satisfactory and modified version of the manuscript acceptable for a publication.

In the results, you present the concentration of NO2 and NO3 approx. 1 mM what correspond to pH approx. 3. However, you declare the pH as 7.5 (line 107), indicating the alkalic solution. Please explain this.

Thank you for this comment. You are right that exposure of water solutions to air plasmas usually leads to their acidification. For example, exposure of deionized water or physiological solution to transient spark (TS) discharge for 10 min leads to a pH decrease to ~ 2.7-3.5. The acidification of the solutions is associated with formation of various species at gas-liquid interface and their dissolution into water solution. Gaseous NO and NO2 dissolve and react with water producing nitrites NO2‾, nitrates NO3‾ and H+. Also HNO2 and HNO3 produced in gas phase can easily dissolve into solution and contribute to NO2‾ and NO3‾ formation and further decrease of pH. Formations of NO2‾ and NO3‾ in water solution are in general considered as main reactions responsible for PAW acidification, i.e. decrease of pH associated with the release of H+ (H3O+).

On contrary to deionized water and physiological solution, in tap water and buffered solutions the situation is little different. In tap water (just like in this study) and after TS discharge exposure for 10 min pH remained fairly constant (pH ~ 7.5) or changed very mildly due to natural hydrocarbon buffer system of tap water that preserves the pH (Kucerova et al. 2019, 10.1002/ppap.201800131). In phosphate buffer water (PB) or phosphate buffer saline (PBS) solutions similar effect was observed after TS discharge exposure when initial pH (6.9) decreased only slightly to ~ 6.2 (Machala 2013, 10.1002/ppap.201200113). One of our previous papers (Kucerova et al. 2019, 10.1002/ppap.201800131) includes a  comparison of properties of deizonized and tap water exposed by TS discharge and changes in pH and concentrations of selected RONS are described.

In general the acidity of the PAW and its buffering capacity is crucial for chemical reactions in water, concentrations of formed reactive species and their subsequent effects on various biological systems (e.g. seeds and plants). In the present study pH effects is however not an issue as it barely changed during experiments and thus quality of PAW used was almost the same.